**Data Availability Statement:** The EPHIA 2017-2018 public release data is now posted on the ICAP PHIA data website (https://phia-data.icap.columbia.

# Factors associated with unawareness of HIV-positive status in urban Ethiopia: Evidence from the Ethiopia population-based HIV impact assessment 2017-2018

Sileshi Lulseged[1][☉][*], Wudinesh Belete[2][☉], Jelaludin Ahmed[3][‡], Terefe Gelibo[1][☉], Habtamu Teklie[2][☉], Christine W. West[4][☉], Zenebe Melaku[1][☉], Minilik Demissie[2][☉], Mansoor Farhani[5][☉], Frehywot Eshetu[3][☉], Sehin Birhanu[4][‡], Yimam Getaneh[2][☉], Hetal Patel[4][‡], Andrew C. Voetsch[4][☉], EPHIA Study Team[¶]

1 ICAP in Ethiopia, Mailman School of Public Health, Columbia University, Addis Ababa, Ethiopia, 2 Department of HIV and Tuberculosis Research, Ethiopia Public Health Institute, Addis Ababa, Ethiopia, 3 United States Centers for Disease Control and Prevention, Addis Ababa, Ethiopia, 4 United States Centers for Disease Control and Prevention, Atlanta, Georgia, United States of America, 5 ICAP. Mailman School of Public Health, Columbia University, New York, New York, United States of America

☉ These authors contributed equally to this work.
‡ These authors also contributed equally to this work.
¶ Membership of the EPHIA Study Group is provided in the Acknowledgments.
* sl2883@cumc.columbia.edu

## Abstract

### Background

The HIV epidemic in Ethiopia is concentrated in urban areas. Ethiopia conducted a Population-based HIV Impact Assessment (EPHIA) in urban areas between October 2017 and April 2018 to measure the status of the country's response to the epidemic.

### Methods

We conducted field data collection and HIV testing in randomly selected households using the national, rapid testing algorithm with laboratory confirmation of seropositive samples using a supplemental assay. In addition to self-report on HIV diagnosis and treatment, all HIV-positive participants were screened for a set of HIV antiretroviral (ARV) drugs indicative of the first- and second-line regimens. We calculated weighted frequencies and 95% confidence intervals to assess regional variation in participants' level of unawareness of their HIV-positive status (adjusted for ARV status).

### Results

We interviewed 20,170 survey participants 15–64 years of age, of which 19,136 (95%) were tested for HIV, 614 (3.2%) tested positive, and 119 (21%) of HIV-positive persons were unaware of their HIV status. Progress towards the UNAIDS first 90 target (90% of people living with HIV would be aware of their HIV status by 2020) substantially differed by administrative region of the country. In the bivariate analysis using log binomial regression, three

edu/files#ethiopia). Dataset documentation is available for immediate download and datasets are available upon request by registering for an account and submitting the request form.

**Funding:** This project was conducted using the U.S. President's Emergency Plan for AIDS Relief (PEPFAR) funds obtained though the U.S Center for Disease Control and Prevention (CDC) under the term of cooperative agreement #U2GGH001226. The findings and conclusions in this report are those of the authors and do not necessarily represent the official position of the funding agency.

**Competing interests:** The authors have declared that no competing interests exist.

regions (Oromia, Addis Ababa, and Harari), male gender, and young age (15–24 years) were significantly associated with awareness of HIV positive status. In multivariate analysis, the same variables were associated with awareness of HIV-positive status.

## Conclusion

One-fifth of the HIV-positive urban population were unaware of their HIV-positive status. The number of unaware HIV-positive individuals has a different distribution than the HIV prevalence. National and regional planning and monitoring activities could address this potentially substantial source of undetected HIV infection by increasing HIV testing among young people, men and individuals who do not use condoms.

## Introduction

In 2014, The Joint United Nations Programme on HIV/AIDS (UNAIDS) set the 90-90-90 targets for the year 2020: 90% of people living with HIV would know their HIV status (first 90), 90% of people with diagnosed HIV infection would receive sustained antiretroviral treatment (ART) (second 90), and 90% of people receiving ART would have viral suppression (third 90) [1]. Huge resources have been invested and various innovations employed in HIV testing and treatment programs since the initiative was launched, and much has been achieved. Country progress towards achieving these targets varied over time across regions and by sex and age, although the disparities are most pronounced in the first and second 90s, and more so in the first 90 [2].

Ethiopia, the second-most populous country in Africa, is heavily affected by the HIV epidemic [3]. The 2011 and 2016 Ethiopian Demographic and Health Surveys (EDHS) [4, 5] estimated the national HIV prevalence among women and men (15–49 years) at 1.5% and 0.9%, respectively. HIV prevalence was seven times higher in urban compared to rural areas, 2.9% versus 0.4% [4]. In 2017, the urban population of Ethiopia constituted 20.4% [6]. The EDHS data indicate that there is a remarkable variation in HIV prevalence across geographic and other sub-population groups [3] Despite persistent efforts to control the epidemic, HIV transmission continues, particularly among the urban population, predominantly being transmitted through unprotected heterosexual sex. A series of isolated surveys have provided data concerning HIV knowledge, attitude, and practice (KAP), as well as sexual behaviour. Key drivers of the epidemic include multiple and concurrent sexual partners, low and inconsistent use of male and female condoms, and mobility and labour migration [5].

According to the EDHS 2016, knowledge about HIV did not vary much by background characteristics except for education; those with no education were less likely to be knowledgeable about HIV. However, awareness about HIV-positive status was limited among the general population, and valid epidemiological evidence was scarce [7]. This would affect achievement of first 90 UNAIDS target, thereby affecting antiretroviral treatment (ART), and viral suppression. By the end of 2017, it was estimated that 75% of HIV-positive people knew their status globally [8], but information on awareness was lacking in Ethiopia. We analysed the Ethiopia Population-based HIV Impact Assessment (EPHIA) data collected in 2017/2018 to determine the prevalence of unawareness of HIV-positive status in adults and its variation by region and other potential determinants.

## Materials and methods

### Study setting and population

Ethiopia has an estimated population of 105 million people in 2018 and the country was administratively divided into nine regional states—Tigray, Afar, Amhara, Oromia, Somali, Benishangul Gumuz, SNNPR, Gambella, and Harari, and two city administrations—Addis Ababa and Dire Dawa. EPHIA was conducted in urban areas across the country. The study population included women and men 0–64 years of age, and this analysis on unawareness of HIV-positive status focused on the age group 15–64 years.

### Data collection procedures

The survey used a household-based cross-sectional study design. We conducted HIV testing in selected households using the national rapid diagnostic testing algorithm with laboratory confirmation of seropositive samples using a supplemental assay. Data collection was conducted from October 2017 to April 2018. Qualitative screening for a detectable concentration of antiretroviral (ARV) drugs was conducted on all participants who were HIV-positive [9]. The adult questionnaire was administered to all eligible participants aged 15 years and older during face-to-face interviews using tablets. Demographic, behavioral and clinical data were collected electronically in the field. Self-reported awareness of HIV-positive status was collected through interviewer-administered questionnaires.

### Sampling procedure

EPHIA used a two-stage, stratified cluster sampling design. The sampling frame for first stage sampling was all urban enumeration areas (EAs) in the country, based on the 2007 Population and Housing Census [3]. Using a probability proportional to size method, 393 EAs were selected randomly. During the second stage, a sample of households was randomly selected within each EA, using an equal probability method, where the average number of households selected per EA was 30 and the actual number of households selected per cluster ranged from 15 to 60, for a total number of 11,810 households.

### Definition of variables

The variables included in this analysis are selected from the EPHIA dataset based on the literature and other scientific evidence to examine their relationship with unawareness. The dependent (outcome) variable for this study is unawareness of HIV-positive status among HIV-positive respondents 15–64 years of age. The outcome was defined as unaware if the respondent reported being unaware of their HIV-positive status and had no detectable ARVs and defined as aware if the respondent were aware of their HIV-positive status or had detectable ARVs. The independent variables included region, socio-demographic and behavioural risk factors.

### Eligibility for the study

Participants were eligible to participate if they lived or slept in the household the night before the interview. Of 12,618 eligible women and 8,920 eligible men, 96.1% of eligible women and 89.6% of eligible men were interviewed, and among these, 95.2% of women and 93.6% of men also had their blood drawn and tested for HIV.

## Data analysis

Data were analyzed using the sample survey procedures and analysis weights were calculated to account for sample selection probabilities and adjusted for non-response and non-coverage using Jackknife replication method, a method that estimates the variance/standard and bias of a large population using a sample data by involving a leave-one-out strategy of the estimation of a parameter in a dataset [10]. All categorical variables were summarized using frequencies and proportions. We calculated weighted frequencies and 95% confidence intervals to assess the association between selected explanatory factors and unawareness of HIV-positive status. In the bivariate analysis, the associations between participants' unawareness of their current HIV-positive status and the potential explanatory factors were tested using the Chi-square test. Variables with Chi-square P-values of ≤0.10 were included in the log binomial regression model. Crude and adjusted prevalence ratios are estimated using log binomial regression model. Age group, education, marital status and sex were included in the log binomial model as confounders. An independent factor was excluded from the multivariable model when the change in the adjusted log-likelihood ratio was not significant with its addition or removal. For collinear factors assessed by looking at the variance inflation factor (VIF) such as involvement in high-risk sex and having multiple sexual partners, only the variable that improved the model more than the other was included in the multivariable model. The analyses were done using STATA version 14.

## Ethical considerations

The survey protocol, screening forms, refusal forms, referral forms, recruitment materials and questionnaires, consent forms, and digital documentation of consent obtained ethical clearance from the respective institution review boards of the Ethiopian Public Health Institute, Centers for Disease Control and Prevention, and Columbia University. As part of the informed consent procedure, all potential participants were informed that participation was voluntary and that they did not need to disclose personal information, which they were uncomfortable sharing, and that they could withdraw from the survey at any time. Prior to initiation any survey procedures, all potential participants were given a printed copy of the consent form in one of six survey languages depending upon their preference. For illiterate participants, an impartial witness chosen by the participant was involved. Potential participants who did not speak any of the six survey languages were considered ineligible. Respondents who consented to participate for the interview were asked to consent separately for biomarker testing. Written parental/guardian permission was obtained for assenting minors. At each stage of the process, consent was indicated by signing or making a mark on the consent form in the tablet and a printed copy. All participants retained written copies. A designated head of household provided written consent for household members to participate in the survey, after which individual members were rostered during the household interview. Participants aged 15–64 years and emancipated minors aged 13–17 then provided the written consent for an interview and for participation in the biomarker component of the survey, including home-based testing and counselling, with return of HIV-test results. Receipt of tests results was a requirement for participation in the biomarker component. If an individual did not want to receive his or her HIV test result, the individual was considered a refusal and excluded from the survey.

## Results

### Participants' characteristics

There were 20,170 survey participants aged 15–64 years in EPHIA, of which 19,136 (95%) were tested and 614 (3.2%) were HIV-positive. As shown in Table 1, 355 (77.2%) of the HIV-

**Table 1. Demographic, socioeconomic and behavioural characteristics of HIV-positive participants aged 15–64 in urban Ethiopia.**

| Background characteristics | Tested N (%) | HIV-Positive (n) | Weighted % | 95% CI* |
|---|---|---|---|---|
| **Region** | | | | |
| Tigray | 1369 (7.2) | 39 | 6.3 | 4.6–8.6 |
| Afar | 821 (1.3) | 32 | 1.8 | 1.2–2.5 |
| Amhara | 2999 (18.8) | 118 | 25.7 | 21.8–30.0 |
| Oromia | 4510 (33.5) | 149 | 33.3 | 29.0–37.8 |
| Somali | 926 (1.3) | 8 | 0.3 | 0.2–0.7 |
| Benishangul Gumuz | 798 (1.3) | 20 | 1.1 | 0.7–1.7 |
| SNNPR | 2665 (16.1) | 49 | 9.4 | 7.1–12.3 |
| Gambella | 788 (0.6) | 44 | 1.1 | 0.8–1.5 |
| Harari | 697 (0.7) | 32 | 1.0 | 0.7–1.4 |
| Addis Ababa | 2780 (18) | 88 | 18.2 | 14.9–21.9 |
| Dire Dawa | 783 (1.2) | 35 | 1.9 | 1.3–2.6 |
| Total | 19136 (100) | 614 | | |
| **Sex** | | | | |
| Female | 11599 (50.1) | 461 | 67.9 | 63.1–72.3 |
| Male | 7537 (49.9) | 153 | 32.1 | 27.7–36.9 |
| Total | 19136 (100) | 614 | | |
| **Age group** | | | | |
| 15–24 years | 7547 (34.8) | 62 | 8.4 | 6.3–11.1 |
| 25–34 years | 5664 (30.5) | 175 | 26.1 | 22.3–30.2 |
| 35–44 years | 3136 (18.9) | 234 | 39.0 | 34.6–43.6 |
| 45–54 years | 1651 (10.1) | 104 | 20.3 | 16.7–24.3 |
| 55–64 years | 1138 (5.7) | 39 | 6.3 | 4.5–8.9 |
| Total | 19136 (100) | 614 | | |
| **Marital status** | | | | |
| Never married | 7103 (35.6) | 71 | 11.4 | 8.8–14.6 |
| Married or living together | 9418 (52) | 285 | 48.1 | 43.6–52.8 |
| Divorced or separated | 1723 (8.6) | 144 | 21.9 | 18.5–25.8 |
| Widowed | 772 (3.8) | 112 | 18.6 | 15.3–22.3 |
| Total | 19016 (100) | 612 | | |
| **Education level** | | | | |
| No education | 2400 (11.9) | 121 | 20.2 | 16.8–24.2 |
| Primary | 6803 (35.3) | 291 | 49.3 | 44.7–53.9 |
| Secondary | 5488 (28.7) | 141 | 22.6 | 19.0–26.6 |
| More than secondary | 4376 (24.1) | 58 | 7.9 | 5.8–10.6 |
| Total | 19067 (100) | 611 | | |
| **Employment status last 12month** | | | | |
| Did not work | 10955(52.7) | 313 | 49.4 | 44.8–54.0 |
| Worked | 8154 (47.3) | 298 | 50.6 | 46.0–55.2 |
| Total | 19096 (100) | 611 | | |
| **Number of sexual partners last 12 months** | | | | |
| No sexual partner | 3689 (27.7) | 241 | 44.0 | 39.2–48.9 |
| One sexual partner | 8778 (68.2) | 270 | 52.6 | 47.6–57.4 |
| Two or more sexual partners | 497 (4.1) | 25 | 3.5 | 2.2–5.4 |
| Total | 12964 (100) | 536 | | |
| **Condom use at last sex in past 12 months** | | | | |
| Used condom | 793 (6.5) | 83 | 16.3 | 12.8–20.4 |

*(Continued)*

**Table 1.** (Continued)

| Background characteristics | Tested N (%) | HIV-Positive (n) | Weighted % | 95% CI* |
|---|---|---|---|---|
| Did not use condom | 8192 (65.1) | 203 | 39.5 | 34.7–44.4 |
| Had no sex | 3689 (28.4) | 241 | 44.3 | 39.4–49.2 |
| Total | 12674 (100) | 527 | | |
| Age at first sexual encounter | | | | |
| Did not have sex before age 15 | 17735 (95) | 536 | 88.8 | 85.7–91.3 |
| Had first sex before 15 | 1014 (5) | 70 | 11.2 | 8.7–14.3 |
| Total | 18749 (100) | 606 | | |

* Confidence interval.

positive participants were from three regions, Oromia, Amhara, and Addis Ababa. Among HI-positive participants, 67.9% were females and 39% were 35–44 years of age. Nearly one-half (48.1%) of the HIV-positive participants were married or living together, 49.3% reported to have primary education, and 50.6% were formally employed in the past 12 months. Four in ten (39.5%) of HIV- positive participants did not use condom in the past 12 months and 88.8% reported they did not have first sex before 15 years of age.

## Testing history

Of all HIV-positive participants aged 15–64 years, 90.4% had ever been tested for HIV and received their results which varied by region, ranging from 84.1% in Afar to 100% Somali and Benishangul Gumuz Region. Among HIV-positive participants who were ever tested for HIV, one-fifth (20.8%) were tested and received results in the past 12 months (Fig 1).

## Self-reported HIV status

Among HIV-positive participants, 74% (466) self-reported they were HIV-positive, 15% (95% CI: 12.0–19.1) self-reported HIV negative status, and 11% (95% CI: 8.1–14.1) self-reported they had never been tested or never received a result.

Among HIV-positive participants, the highest proportion (25.1%, 95% CI: 14.2–40.4) of self-reported HIV negative participants were from the Gambella region (Fig 2). In Afar, Addis Ababa, and Dire Dawa, more people never tested than self-reported negative, whereas in the rest of the other regions, more people self-reported negative.

## Unawareness of HIV-positive status

Combining self-reported awareness and adjustment of ARV status, among HIV-positive participants aged 15–64 years, 21% (95% CI: 17.3–25.3) were found unaware of their HIV-positive status. A significantly higher proportion of men (30%, 95% CI: 21.9–39.4) were unaware of their HIV-positive status compared to women (16.7%, 95% CI: 13.1–21.0) (Table 2). Unawareness was highest among HIV-positive participants aged 15–24 years. Unawareness among those 15–24 years of age was 37% (95% CI: 24.2–51.9) compared to 14.5% (95% CI: 6.0–30.8) among those 55–64 years; however, the difference was not statistically significant. Unawareness was significantly higher among those who did not use a condom (30%, 95% CI: 22.7–38.6), compared to those who used a condom in the last sexual encounter in the past 12 months (8.7%, 95% CI: 4.0–17.8). Unawareness was 28.1% (95% CI: 21.3–36.2) among male headed households versus 16.4% (95% CI: 4.0–17.8) among female headed households.

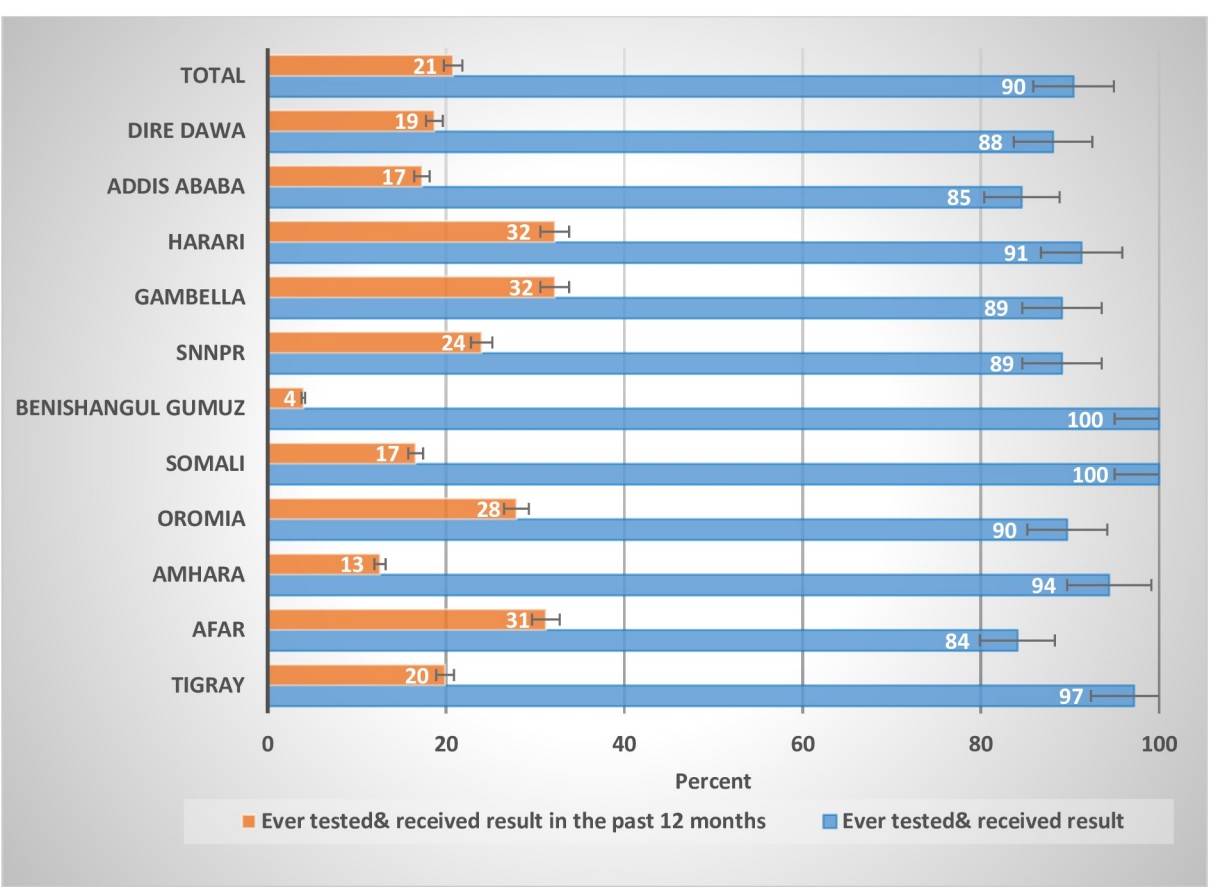

**Fig 1. Distribution of self-reported HIV testing coverage by region among HIV-positive participants aged 15–64 years, 2017–18.**

Unawareness was 22.1% (95% CI: 18.0–26.8) among those who reported first sex at or after 15 years compared to 11.2% (95% CI: 5.5–21.5) among those who stated first sex before 15 years. Among those who were unaware of their HIV status, 84% had never been tested and 14% had been tested for HIV, a difference that was significant (p = 0.0001). Unawareness was 79.3% (95% CI: 71.2–85.6) among those who had no ARV detected in their blood (Table 2).

The level of unawareness varied across the regions, ranging from zero in the Somali region to 33.4% in Gambella (Table 2). Progress towards the first 90 target substantially differed by region in urban Ethiopia, where the highest burden (78%) were from two most populous regions (Amhara, Oromia) and Addis Ababa, the capital city, while the lowest burden (1.8%) was in Gambella (a small region), though the region had the highest HIV prevalence (Fig 3).

## Factors associated with unawareness of HIV-positive status

In a bivariate analysis using the log binomial regression, being from Afar, Oromia, SNNPR, Gambella, Harari, or Addis Ababa region, age 15–24, male gender, primary education level, male headed household, not using condom in the last sexual encounter in the past 12 months, and age less than 15 at first sexual encounter were significantly associated with unawareness of HIV-positive status. In the multivariable log binomial regression model, education level, gender of head of the household and age at first sex were not significantly associated with unawareness (Table 3).

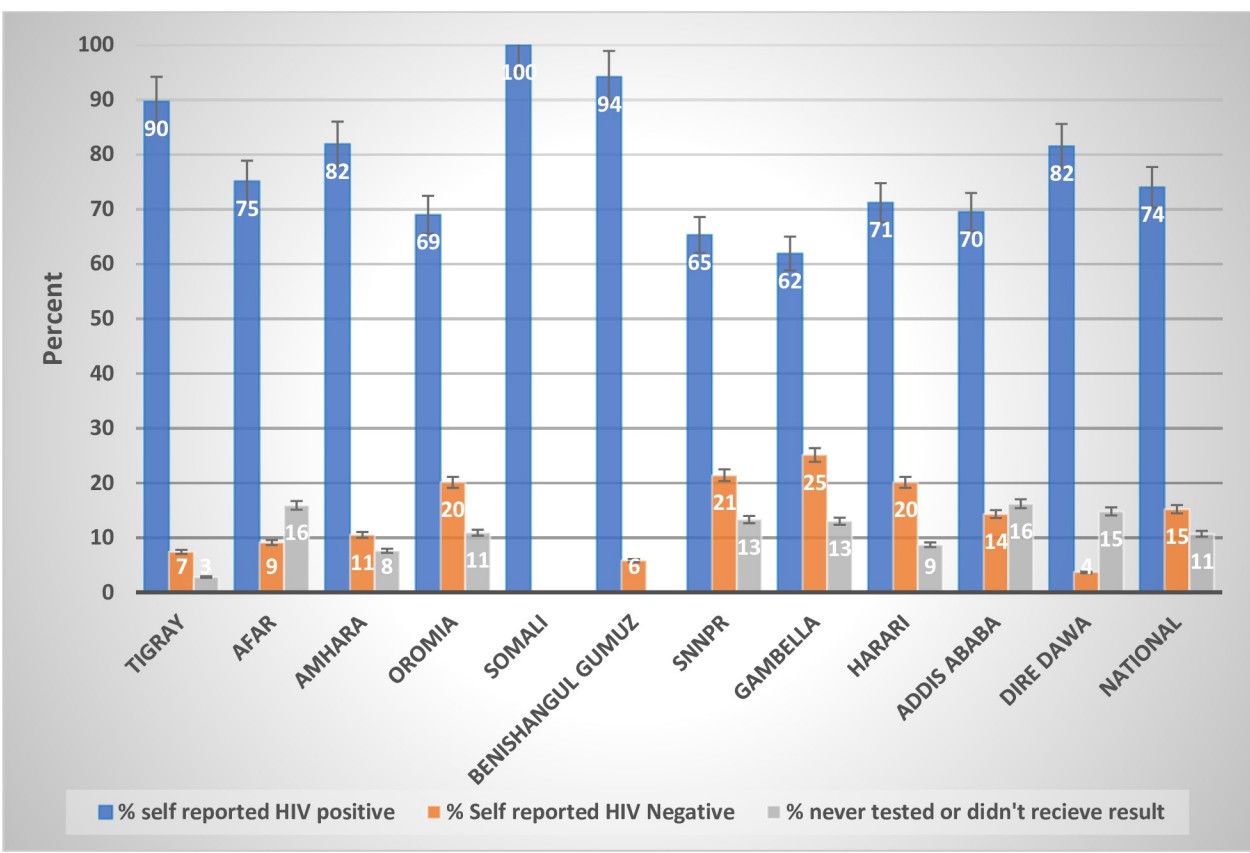

**Fig 2. HIV-positive status by history among HIV-positive participants aged 15–64 years in urban Ethiopia, EPHIA 2017–2018.**

Table 3 provides the results of a log binomial regression analysis. Some demographic and behavioral factors were independently associated with higher probability of unawareness. After controlling for marital status, educational level, age and sex using multivariable log bino-mial regression analysis, three regions (Oromia, Addis Ababa and Harari), male gender, age group (15–24 years) and condom non-use in the past 12 month were significantly associated with unawareness of HIV positive status. Three regions Oromia (APR 1.9, 95% CI: 1.3–2.8), Harari (APR 2.2, 95% CI: 1.1–4.4) and Addis Ababa city administration (APR 2.2, 95% CI: 1.3–3.8) had higher prevalence of unawareness as compared with Benishangul Gumuz region. The probability of HIV-positive status unawareness was higher among males (APR 1.3, 95% CI: 1.2–2.7) compared to females.

There was a greater prevalence of being unaware of HIV-positive status among young peo-ple in the age group 15–24 years (APR1.8, 95% CI: 1.3–3.0). The prevalence of unawareness was higher among individuals who did not use condom in the past 12 months (APR 2.8, 95% CI: 1.4–6.0) compared with those who used condom during the last sexual encounter in the past 12 months.

## Discussion

We found more than one-fifth (21%) of HIV-positive infected participants were unaware of their HIV status, which is less than the 25% global estimate in 2017 [11]. However, the

**Table 2. ARV adjusted estimates of unawareness by socio demographic and behavioral characteristics among HIV-positive participants aged 15–64 years, EPHIA 2017/2018.**

| Characteristics | HIV-Positive (N) | Unaware | | 95% CI* | P-value |
|---|---|---|---|---|---|
| | | n | % | | |
| **Region** | | | | | |
| Tigray | 38 | 4 | 11 | [3.9–25.3] | 0.138 |
| Afar | 32 | 8 | 25 | [12.8–42.9] | |
| Amhara | 118 | 15 | 15 | [8.8–23.8] | |
| Oromia | 148 | 29 | 24 | [17.0–32.8] | |
| Somali | 8 | - | 0 | - | |
| Benishangul Gumuz | 20 | 1 | 5.8 | [0.8–31.7] | |
| SNNPR | 49 | 13 | 27 | [15.9–41.1] | |
| Gambella | 44 | 14 | 33 | [20.7–49.1] | |
| Harari | 32 | 9 | 27 | [14.3–44.5] | |
| Addis Ababa | 86 | 21 | 25 | [16.8–36.0] | |
| Dire Dawa | 34 | 5 | 16 | [6.8–33.7] | |
| **Sex** | | | | | |
| Female | 456 | 78 | 17 | [13.1–21.0] | 0.003 |
| Male | 153 | 41 | 30 | [21.9–39.4] | |
| **Age group** | | | | | |
| 15–24 years | 62 | 23 | 37 | [24.2–51.9] | 0.135 |
| 25–34 years | 172 | 31 | 20 | [13.3–29.2] | |
| 35–44 years | 233 | 43 | 21 | [14.9–28.1] | |
| 45–54 years | 104 | 17 | 18 | [10.9–27.6] | |
| 55–64 years | 38 | 5 | 15 | [6.0–30.8] | |
| **Marital status** | | | | | |
| Never married | 71 | 18 | 24 | [14.2–37.0] | 0.461 |
| Married or living together | 285 | 57 | 23 | [17.4–29.9] | |
| Divorced or separated | 140 | 26 | 20 | [12.8–28.9] | |
| Widowed | 111 | 17 | 15 | [8.9–24.5] | |
| **Education level** | | | | | |
| No education | 120 | 19 | 14 | [8.0–22.8] | 0.165 |
| Primary | 288 | 58 | 24 | [18.1–30.5] | |
| Secondary or higher | 198 | 41 | 21 | [14.8–28.4] | |
| **Religion** | | | | | |
| Muslim | 82 | 16 | 20 | [10.7–34.2] | 0.911 |
| Christian | 522 | 99 | 21 | [16.7–25.2] | |
| **Wealth quintile** | | | | | |
| Lowest | 102 | 20 | 26 | [15.6–38.8] | 0.286 |
| Second | 107 | 17 | 13 | [7.0–23.5] | |
| Middle | 142 | 23 | 18 | [11.5–26.7] | |
| Fourth | 145 | 33 | 25 | [17.2–33.6] | |
| Highest | 113 | 26 | 24 | [15.9–34.0] | |
| **Gender of household head** | | | | | |
| Female headed | 396 | 67 | 16 | [12.5–21.1] | 0.005 |
| Male headed | 213 | 52 | 28 | [21.3–36.2] | |
| **Condom use at last sex in past 12 months** | | | | | |
| Used condom | 83 | 8 | 8.7 | [4.0–17.8] | 0.001 |
| Did not use condom | 201 | 51 | 30 | [22.7–38.6] | |

*(Continued)*

**Table 2.** (Continued)

| Characteristics | HIV-Positive (N) | Unaware | | 95% CI* | P-value |
|---|---|---|---|---|---|
| | | n | % | | |
| No sex in the past 12 month | 238 | 39 | 18 | [12.6–24.6] | |
| **Age at first sexual encounter** | | | | | |
| First sex 15 + years | 532 | 105 | 22 | [18.0–26.8] | 0.047 |
| First sex before 15 | 69 | 11 | 11 | [5.5–21.5] | |
| **Ever been tested for HIV** | | | | | |
| Never tested | 51 | 44 | 84 | [69.4–92.4] | 0.0001 |
| Ever tested | 556 | 73 | 14 | [10.9–18.2] | |
| **ARV detected in their blood** | | | | | |
| No | 150 | 119 | 79.3 | [71.2–85.6] | 0.0001 |
| Yes | 459 | 0 | 0 | | |
| Total | 609 | 119 | 21 | [17.3–25.3] | |

* Confidence interval.

unawareness level identified in EPHIA is based on urban population and the EPHIA plausibility interval (95% CI, 17.3–25.3) covers the global estimate of 25%. In Ethiopia, a country with huge population size, one-fifth of which is urban population [6], where HIV prevalence is seven times higher in urban areas than in rural areas (2.9% versus 0.4%) [12], being unaware

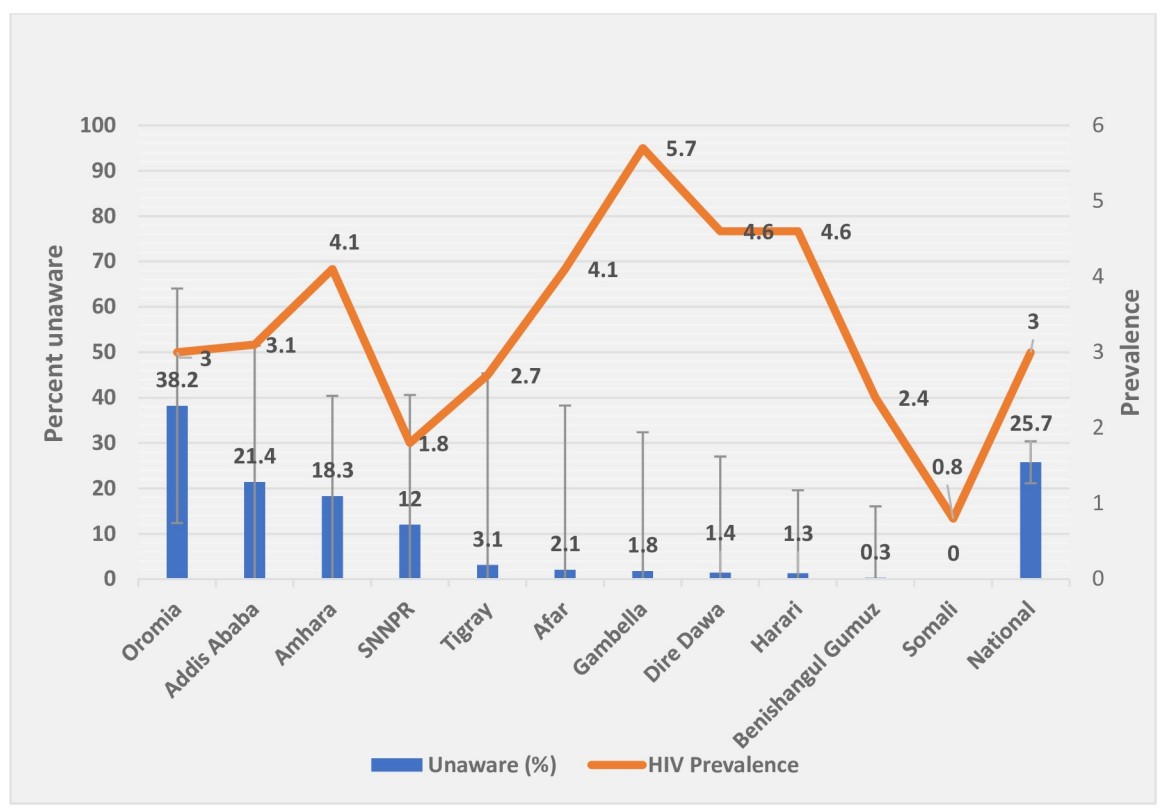

**Fig 3. The burden of unawareness status by region among HIV-positive participants aged 15–64 in urban Ethiopia, 2017/18.**

**Table 3. Factors associated with weighted unawareness status among HIV-positive participants aged 15–64 years, EPHIA 2017/2018.**

| Characteristics | CPR* (95% CI) | P-value | APR** (95% CI) | P-value |
|---|---|---|---|---|
| **Region[§]** | | | | |
| Benishangul Gumuz | 1 | | 1 | |
| Tigray | 1.8 (0.4–8.06) | 0.439 | 0.7 (0.2–2.21) | 0.507 |
| Afar | 4.3 (1.08–16.91) | 0.039 | 1.5 (0.68–3.45) | 0.301 |
| Amhara | 2.5 (0.6–10.7) | 0.205 | 1.2 (0.68–2.26) | 0.495 |
| Oromia | 4.1 (1.09–15.44) | 0.036 | 1.9 (1.25–2.77) | 0.002 |
| SNNPR | 4.6 (1.16–17.99) | 0.030 | 1.4 (0.74–2.7) | 0.291 |
| Gambella | 5.7 (1.42–23.07) | 0.014 | | |
| Harari | 4.6 (1.13–18.63) | 0.034 | 2.2 (1.06–4.39) | 0.034 |
| Addis Ababa | 4.3 (1.16–16.09) | 0.029 | 2.2 (1.27–3.83) | 0.005 |
| Dire Dawa | 2.8 (0.63–12.1) | 0.178 | 1.3 (0.51–3.32) | 0.574 |
| **Sex** | | | | |
| Female | 1 | | 1 | |
| Male | 1.8 (1.3–2.49) | 0.0001 | 1.3 (1.22–2.69) | 0.009 |
| **Age group** | | | | |
| 15–24 years | 2.5 (1.09–5.99) | 0.031 | 1.8 (1.32–2.98) | 0.003 |
| 25–34 years | 1.3 (0.58–3.33) | 0.458 | 1.3 (0.68–4.85) | 0.238 |
| 35–44 years | 1.4 (0.66–3.15) | 0.362 | 0.7 (0.61–4.15) | 0.07 |
| 45–54 years | 1.2 (0.52–2.92) | 0.636 | 0.6 (0.38–2.0) | 0.56 |
| 55–64 years | 1 | | 1 | |
| **Education level** | | | | |
| No education | 1 | | 1 | |
| Primary education | 1.7 (1.02–2.89) | 0.041 | 1.2 (0.7–2) | 0.537 |
| Secondary and above education | 1.3 (0.74–2.34) | 0.355 | 1.1 (0.6–2) | 0.715 |
| **Condom use at last sex in past 12 months** | | | | |
| Used condom | 1 | | | |
| Did not use condom | 3.4 (1.39–8.54) | 0.008 | 2.8 (1.4–6) | 0.006 |
| No sex in the past 12 month | 2.0 (0.89–4.65) | 0.089 | 1.2 (0.8–1.7) | 0.371 |

\* Crude prevalence ratio

\*\*Adjusted prevalence ratio.

[§] Somali region was not included in the model due to a small number of cases.

of their HIV-positive status indicates that a substantial number of HIV-positive people are not getting tested, seeking care and treatment and remain a potential source of transmission of HIV infection. The HIV-positive status unawareness reported in this study is lower than that reported in other countries in the region, 52% in Uganda [13], 39% in Mozambique [14], 38.3% in Malawi [15], 44.0% in Zambia [16], 32.9% in Tanzania [17] and 62.8% in Cote d'Ivoire [18]. This difference could partly be explained by the difference in the source and study population, including our study, which focused on urban population, where there was a better access to information, care and treatment services than rural settings. As reported by Deribew et al. [19], the overall capacity score for HIV diagnosis and treatment, which was estimated based on the assessment of structure, process and overall capacity framework constructed by taking the average of all indicators and rescaling it to 100, was higher in urban facilities (57.1%) than that of the rural health facilities (38.2%). The prevalence of unawareness in this study was higher than the finding reported in Kenya 16.2% [20, 21], 16.2% in Rwanda

[22], and 15% in South Africa [23] and almost comparable with another report from Kwa-Zulu-Natal province in South Africa (24.8%) [24].

As our findings suggest that 15.2% of HIV-positive persons self-reported not to be infected based on their last HIV test, a potential respondent bias could have contributed to the difference. Detailed analysis of data from household surveys in Africa indicated that, even after adjusting for expected seroconversions, one-quarter to one-third of HIV positive respondents intentionally misreported their HIV-positive status as negative [25, 26]. Raymond, et al. [27] highlighted the gaps in HIV diagnosis, which might be unattainable under the ambitious UNAIDS 90–90–90 targets given the current trends. This could explain most of the observed variation in addition to methodological differences. No single method may be fully effective to increase progress towards the first 90, but when they are used in combination and supported by structural changes they could be more effective [8]. Studies in Ethiopia also indicate that Voluntary Counselling and Testing (VCT) utilization varies by geographic regions of the country in both men and women [28].

Based on our findings, distinguishing between the burden of unaware population and the prevalence of HIV stratified by administrative regions could be useful when designing programs and targeting interventions. The progress towards the first 90 target substantially differed by region in urban Ethiopia. Based on our survey, over three-quarters of those unaware of their HIV- positive status were from three regions (Oromia, Amhara, and Addis Ababa), posing a barrier to HIV prevention, care, and treatment efforts in country (Table 2). Of the estimated 79,827 unaware adult HIV-positive population in urban Ethiopia, 62,183 (77.9%) were from these three regions. In contrast, though Gambella was the region with highest prevalence, among all HIV-positive individuals who were unaware of their HIV-positive status, only 1,443 (1.8%) were those unaware of their HIV-positive status. Identification and meeting administrative region-specific situations could help in finding those with undetected infection and being on track to end AIDS as a pandemic by 2030. These and measures like improving availability of testing service, which currently stands at 70%-76.6% [19] could help in HIV case identification.

The adaptation of community directed intervention approaches used elsewhere [29] and those that are culturally appropriate and less costly in low resource communities can be considered as an alternative strategy to expand coverage. For example, expanding the population coverage of evidence-based interventions with health extension workers such as door-to-door HIV testing during the provision of home visit services. Identification and working with community structures have shown to promote trust, equity and respect, and enhance delivery of essential services to every eligible member of the community [29]. The availability of HIV testing services is currently lower (45%) in Gambella than the national average (74.5%) [19]; this might also contribute to the low awareness of HIV-positive status.

A region-specific analysis showed that the highest proportion of people who were unaware of their HIV-positive status was from Gambella, the region with the highest HIV prevalence. An increase in HIV-positive status unawareness among the population has an impact on increased HIV infection. Consequently, the high percentage of unaware population in Gambella region might have contributed to the high prevalence of HIV in that region. The high rates of undiagnosed infection may suggest limited coverage of testing services and a relatively higher incidence. A study in Ethiopia indicated that HIV testing service was available in 74.5% of health facilities, which varied by region, ranging across facilities from 44.4% in Benishangul Gumuz to 88% in Tigray and Afar [19].

Studies elsewhere have shown that the HIV transmission rate among persons unaware of their HIV- positive status was three to seven times higher than the rate among those aware of their status [30]. Others have reported that for every percentage point increase in HIV-positive

status awareness, the HIV incidence in monogamous populations decreases by 0.27% for women and 0.63% for men [31], indicating a HIV risk increase with level of unawareness. These are useful observations and need to be elaborated through modelling analysis in the future.

Knowledge of one's HIV positive status presumably leads to behaviour change and results in HIV- positive people taking measures to reduce the spread of the virus to uninfected persons. In Gambella, targeted interventions involving sexually active men and women could reduce transmission from persons who were unaware of their HIV seropositivity. It could also help to convey the opportunity to increase Voluntary Medical Male Circumcision (VMMC) in Gambella in the context of the high prevalence of uncircumcised men there. Administrative regions could consider their specific sociodemographic and behavioural factors in the planning for interventions targeting the first 90 and in monitoring progress towards the 2020 goals.

Three regions, Oromia (APR 1.9, 95% CI: 1.3–2.8), Harari (APR 2.2, 95% CI, 1.1–4.4) and Addis Ababa (APR 2.2, 95% CI: 1.3–3.8) had higher prevalence of unawareness as compared with Benishangul Gumuz region. Studies revealed that there was significant difference in the uptake of VCT by region, which partly reflect the multicultural characteristics of the country and difference in the pace of implementation of the health extension program (HEP) [28]. There were significant regional disparities in ART coverage as well, 63% in Amhara and 43% in Oromia [19].

Male gender was associated with increased unawareness among HIV-positive participants (APR 1.3, 95% CI: 1.2–2.7). A higher prevalence of HIV-positive status unawareness among male respondents in our survey was consistent with study reports from South Sudan [8] and Uganda [13], which documented that women were more likely to report that they knew their HIV-positive status than their male counterparts [32]. Similarly, in Mozambique, men had twice the odds of being unaware of their serostatus compared with women [14]. Another study also indicated that unawareness was more common among men than women, 32.7% vs 22.3% [24]. A study in Uganda indicated that females had a 1.26 times higher odds of awareness of HIV-positive status than males [13]. The gender difference in awareness of HIV-positive status could be due to efforts made to increase HIV testing and counselling, which might have benefited women more than men in accessing services. Integration of HIV and antenatal services affords an opportunity for women of childbearing age to access routine HIV testing. The higher level of unawareness of HIV-positive status among adult males suggests a need to utilize alternative HIV testing approaches for this group. As shown elsewhere in Sub-Saharan Africa, door-to-door HIV testing and counselling may be an option for increasing access to testing for male adults in Ethiopia [33]. The other option could be self-testing. Pregnant women and lactating mothers who test positive at antenatal care and mother-and-child centres are provided self-testing kits to give to their male sexual partners to know their status [34].

As in Mozambique [14], HIV-positive people who reported not using a condom during their last sexual intercourse were more likely to be unaware of their HIV-positive status in our study. Unawareness of HIV-positive status was associated with non-condom use in the past 12 months in urban Ethiopia (APR 2.8, 95% CI: 1.4–6.0). Risky sexual behavior seems to be associated with factors linked to poor health-seeking behavior, which may have negative implications for HIV testing and treatment as well as prevention [24]. A previous study reported that perceived low risk of HIV infection is a major barrier to uptake of HIV testing and may undermine the benefits of increasing ART availability in sub-Saharan Africa [35]. Individuals often assume that they are at low risk of infection if they are currently abstinent, have a steady partner, are not part of a high-risk group, or do not have physical symptoms of illness.

Among the sociodemographic and economic factors, age was significantly associated with unawareness. Young people aged 15–24 years had a significantly higher prevalence of unawareness than the older adults (p = 0.003). This is consistent with other studies, where young people were more likely to be unaware than older groups [24]. Educational level and age at first sexual encounter were associated with HIV-positive status unawareness in a bivariate analysis, but we could not demonstrate an independent association for these variables. Wealth quintile was not associated with unawareness of HIV-positive status. However, studies indicated that HIV-positive people from households in the richest wealth quintile were more likely to be aware of their HIV- positive status than those in the middle wealth quintile [14]. In urban Ethiopia, economic inequalities may be minimal among the survey participants.

## Limitations

Our study covers urban areas of Ethiopia and there are socio-economic and behavioural factors that were not controlled for in our study. Some regions such as Somali and Benishangul Gumuz had a relatively small number of HIV-positive people, which may raise questions related to accuracy of unawareness estimates in these regions. This study also has had the inherent limitation of a cross-sectional study design, which does not allow examining cause and effect relationships. Furthermore, the survey was conducted only in urban areas and might miss the full effect of population dynamics in the country, including mobility, migration, and transmission risks in the rural population.

## Conclusions

As shown by our study, Ethiopia was lagging behind the UNAIDS first 90 target by 2018. There was a significant variation in HIV-positive status unawareness by region, male gender, and young age and HIV risk factors such as condom non-use. The high rates of undiagnosed infection may suggest limited coverage of testing services and relatively high incidence. The number of unaware HIV-positive individuals has a different distribution than the HIV prevalence or percent unaware, which is a critical distinction to control the epidemic. The results of this study can be used to inform how administrative regions use available evidence to make program decisions. The national program could improve the HIV testing programs in increasing awareness among men, individuals who do not use condom and those 15–24 years of age. Further analysis on the level of unawareness of HIV-positive status and service uptake are needed to better understand how individual, community and structural factors contribute to the regional variation.

## Supporting information

**S1 File. The EPHIA study team.**
(DOCX)

## Acknowledgments

We would like to extend our thanks to the leadership at the Ministry of Health, EPHI, the regional health bureaus (RHBs) and their sub-regional units, CDC and ICAP for their administrative support in organizing and conducting the survey. Our thanks also go to field coordinators, supervisors and data collectors for their dedicated work and all study participants for providing the necessary information.

## The EPHIA study team

A list of the study team (survey investigators) and is available from: https://phia.icap.columbia.edu/wp-content/uploads/2020/11/EPHIA_Report_280820_High-Res.pdf; and also uploaded as Supporting Information.

Lead author of the group: Andrew C. Voetsch

E-mail: aav6@cdc.gov

## Author Contributions

**Conceptualization:** Sileshi Lulseged, Wudinesh Belete, Jelaludin Ahmed, Terefe Gelibo, Habtamu Teklie, Christine W. West, Zenebe Melaku, Minilik Demissie, Mansoor Farhani, Frehywot Eshetu, Sehin Birhanu, Yimam Getaneh, Andrew C. Voetsch.

**Data curation:** Sileshi Lulseged, Jelaludin Ahmed, Terefe Gelibo, Zenebe Melaku, Mansoor Farhani, Frehywot Eshetu, Yimam Getaneh, Andrew C. Voetsch.

**Formal analysis:** Sileshi Lulseged, Jelaludin Ahmed, Terefe Gelibo, Christine W. West, Mansoor Farhani, Yimam Getaneh, Andrew C. Voetsch.

**Funding acquisition:** Andrew C. Voetsch.

**Investigation:** Sileshi Lulseged, Wudinesh Belete, Jelaludin Ahmed, Terefe Gelibo, Habtamu Teklie, Christine W. West, Zenebe Melaku, Minilik Demissie, Mansoor Farhani, Frehywot Eshetu, Sehin Birhanu, Yimam Getaneh, Hetal Patel, Andrew C. Voetsch.

**Methodology:** Sileshi Lulseged, Wudinesh Belete, Jelaludin Ahmed, Terefe Gelibo, Habtamu Teklie, Christine W. West, Zenebe Melaku, Minilik Demissie, Mansoor Farhani, Frehywot Eshetu, Sehin Birhanu, Yimam Getaneh, Hetal Patel, Andrew C. Voetsch.

**Project administration:** Sileshi Lulseged, Zenebe Melaku, Mansoor Farhani, Yimam Getaneh, Andrew C. Voetsch.

**Resources:** Sileshi Lulseged, Zenebe Melaku, Andrew C. Voetsch.

**Software:** Sileshi Lulseged, Jelaludin Ahmed, Terefe Gelibo, Christine W. West, Zenebe Melaku, Mansoor Farhani, Andrew C. Voetsch.

**Supervision:** Sileshi Lulseged, Wudinesh Belete, Jelaludin Ahmed, Terefe Gelibo, Habtamu Teklie, Zenebe Melaku, Minilik Demissie, Mansoor Farhani, Frehywot Eshetu, Sehin Birhanu, Yimam Getaneh, Hetal Patel, Andrew C. Voetsch.

**Validation:** Sileshi Lulseged, Jelaludin Ahmed, Terefe Gelibo, Zenebe Melaku, Mansoor Farhani, Frehywot Eshetu, Yimam Getaneh, Andrew C. Voetsch.

**Visualization:** Sileshi Lulseged, Jelaludin Ahmed, Terefe Gelibo, Christine W. West, Zenebe Melaku, Mansoor Farhani, Frehywot Eshetu, Yimam Getaneh, Andrew C. Voetsch.

**Writing – original draft:** Sileshi Lulseged, Wudinesh Belete, Jelaludin Ahmed, Terefe Gelibo, Habtamu Teklie, Christine W. West, Zenebe Melaku, Minilik Demissie, Mansoor Farhani, Frehywot Eshetu, Sehin Birhanu, Yimam Getaneh, Hetal Patel, Andrew C. Voetsch.

**Writing – review & editing:** Sileshi Lulseged, Wudinesh Belete, Jelaludin Ahmed, Terefe Gelibo, Habtamu Teklie, Christine W. West, Zenebe Melaku, Minilik Demissie, Mansoor Farhani, Frehywot Eshetu, Sehin Birhanu, Yimam Getaneh, Hetal Patel, Andrew C. Voetsch.

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
