## [Decision Letter · Decision Letter 0]

14 May 2021

PONE-D-21-05302

Factors associated with Unawareness of HIV Positive Status in Urban Ethiopia: Evidence from the Ethiopia Population-based HIV Impact Assessment 2017-2018

PLOS ONE

Dear Dr. Lulseged,

Thank you for submitting your manuscript to PLOS ONE. After careful consideration, we feel that it has merit but does not fully meet PLOS ONE’s publication criteria as it currently stands. Therefore, we invite you to submit a revised version of the manuscript that addresses the points raised during the review process.

There was a change of editor, and one high-quality review on the research is available that suggests that the paper can fulfill PLOS ONE publication criteria with minor changes. This is also the opinion of the new academic editor. In addition to the detailed comments by the reviewer please take note of:

Data availability statement: You argue that “No - some restrictions will apply” regarding data availability but “All relevant data are within the manuscript and its Supporting Information files.”. Those two statements are not compatible. Please following PLOS data policy provide the URL of the public repository and the conditions of access.PLOS ONE being a multidisciplinary journal written for a general audience, the authors should not take for granted knowledge of what the 90-90-90 targets are as in line 68. That should be provided as background stating that the paper is concerned about the first 90. Instead, it does not to be fully described later in the results (l. 209).A third concern is the way that unawareness is measured as a proportion of all HIV+. As the reviewer states, the policy implications of this research is to look for those unaware of their HIV+ status. While this focus is not wrong and corresponds to the first 90 target, it should go together with the complementary perspective: the proportion HIV+ unaware people among those declaring not to be HIV+, since this has implications regarding testing costs and cost-benefit of a testing programme. There is also no information regarding whether HIV+ unaware people had previously tested. Please provide at least descriptive measures in these respects, maybe a table similar to table 1 for those HIV+ unaware as proportion of those declaring to be HIV and some tabulation of those unaware HIV+ according to previous testing. This is a concern due to the effects that testing can have on behaviour, please check Gong, E. (2015), “HIV Testing and Risky Sexual Behaviour”. *Econ J*, 125: 32-60. https://doi.org/10.1111/ecoj.12125.

We look forward to receiving your revised manuscript.

Kind regards,

José Antonio Ortega, Ph.D.

Academic Editor

PLOS ONE

Journal Requirements:

3) Please provide additional details regarding participant consent. In the ethics statement in the Methods and online submission information, please ensure that you have specified (1) whether consent was informed; (2) whether the ethics committee(s) approved digital documentation of consent; (3) whether parents/guardians provided written informed consent for minors to participate in the study, or whether the ethics committee(s) waived the need for this.

4)  Thank you for stating the following in the Acknowledgments Section of your manuscript:

[This project was conducted using the U.S. President’s Emergency Plan for AIDS Relief (PEPFAR) funds obtained though the U.S Center for Disease Control and Prevention (CDC) under the term of cooperative agreement #U2GGH001226. The findings and conclusions in this report are those of the authors and do not necessarily represent the official position of the funding agency..]

 [The funders had no role in study design, data collection and analysis, decision to

publish, or preparation of the manuscript.]

5) One of the noted authors is a group or consortium [EPHIA Study Team]. In addition to naming the author group, please list the individual authors and affiliations within this group in the acknowledgments section of your manuscript. Please also indicate clearly a lead author for this group along with a contact email address.

Reviewers' comments:

Reviewer's Responses to Questions

**Comments to the Author**

1. Is the manuscript technically sound, and do the data support the conclusions?

Reviewer #1: Yes

2. Has the statistical analysis been performed appropriately and rigorously? 

Reviewer #1: Yes

3. Have the authors made all data underlying the findings in their manuscript fully available?

Reviewer #1: Yes

4. Is the manuscript presented in an intelligible fashion and written in standard English?

Reviewer #1: Yes

5. Review Comments to the Author

Reviewer #1: Reviewer Report for “Factors associated with Unawareness of HIV Positive Status in Urban Ethiopia: Evidence from the Ethiopia Population-based HIV Impact Assessment (2017-2018)”

Overall Comments:

The paper focuses on people who are HIV-positive and unaware of their status, broken down by different epidemiologic factors. Overall, this is an important study in guiding future testing and treatment efforts. In that vein, one of the strongest claims in the paper is that future efforts should focus on where there are more people living with HIV who are unaware of their status (Addis Ababa, Harari, Oromio), rather than where there is high prevalence (Gambella). However, the paper loses focus toward the end as the authors expound on every finding rather than only the main claims. There are several limitations to the study including that only urban enumeration areas which the authors state is 20% of the population, and possibly that ARV testing was only done for participants who reported being HIV-positive. Despite these limitations, the study uses national data to draw conclusions on how to increase HIV awareness. This paper makes an important contribution to the literature and help guide future HIV testing and treatment policies in Ethiopia.

Major Compulsory Revisions

Abstract:

Line 18: This line implies that all people who tested positive for HIV were also screened for ARVs, whereas later (Line 87), it is implied that only individuals who self-report HIV positive status were screened for ARVs. Please clarify.

Also, throughout the paper, the terms “HIV positive,” “HIV positive status,” “HIV+,” “HIV infected,” and “PLHIV” are used. Unclear to me if this is intentional, but would benefit from consistency.

Results:

Line 188: I think ARV data could be a very important aspect of this study. Was ARV testing only done for participants who reported being HIV-positive? Readers would be very interested in the data on ARV testing among all HIV-positive individuals. It would change our interpretation of the positivity data as well as risk factors such as condom use.

Line 203: Please clarify the column headings in Table 2. Three of them are labeled “(N)” and it’s unclear what are the numerators and denominators for the “%” column.

Line 214: It’s unclear to me what the left y-axis of this graph is. I would assume it is percentage of people who tested HIV-positive who are unaware of their status, but these numbers are different from those in Table 2.

Minor Essential Revisions

Abstract:

Line 23: Unclear to me why a percentage isn’t placed after the number tested HIV positive (3.2%). Is it due to the sampling strategy?

Line 34: “One fifth of the urban population were” should be “One-fifth of the HIV-positive urban population were.”

Introduction:

Lines 62-63: The first estimate of 75% refers to PLHIV knowing their HIV status, whereas the second numbers of 97% and 99% refer to knowledge of HIV. I’d consider either changing this to make the numbers more consistent or focus on framing each number to explain the train of logic.

Line 68: You mention the “first 90 UNAIDS and country targets” but I don’t think the targets have been clearly introduced yet.

Methods:

Line 77: Missing a comma between Afar and Amhara.

Line 78: Do not need a comma between “Addis Ababa” and “and Dire Dawa”

Line 77-78: Consider using an m-dash (—) instead of colons given the use of two colons in the sentence.

Line 105: There is an extra space between “HIV” and “+”

Line 114: “Data was analyzed using” should be changed to “Data were analyzed using”

Line 122: Please clarify what is meant by “priority confounders.”

Line 124: How was collinearity assessed?

Line 124: “was not significant on its” should be changed to “was not significant with its”

Line 125: Please clarify what is meant by “only the variable that improve the model was included.” I would assume that if two variables are collinear and one improves the model, then would both improve the model. Presumably one could improve the model more than the other.

Results:

Line 145: Is there a reason no percentage is placed after the number HIV positive?

Lines 146-147: “Among HIV positive participants, females constituted” should be changed to “Among HIV positive participants, 67.9% were females and 39% were”

Line 164: There is some inconsistent bolding in the table. The word “sex” in “Had no sex” is bolded. “N=606” is not bolded. Two words in “Did not have sex before age 15” appear bolded.

Line 166: The sub-heading “Self-reported HIV Status” does not have a colon at the end. Throughout the paper, some sub-headings have a colon and others do not.

Line 170: In Figure 1, is there a reason why Somali has a greater proportion that has tested and received results in the last 12 months than the proportion that ever tested and received results? I’m guessing this is due to incomplete data but should clarify.

Line 173: Why does this figure only include ages 25-64 when all others are 15-64?

Line 177: “74% self-reported they were HIV” should be changed to “74% self-reported they were HIV-positive”

Line 178: “11% reported they had” should read “11% self-reported they had”

Line 180: I think in general, the order of parentheses and brackets should be with parentheses on the outside and brackets on the inside. This is done multiple times throughout the paper.

Line 189: “ARV status, HIV positive” should read “ARV status, among HIV positive”

Line 192: Some of the brackets/parentheses are mixed up in this line.

Line 203: The last row presumably is the total among everyone. This row could be labeled more obviously or be closer to the top of the table.

Line 218: Table 3 actually shows quite a few other variables that are associated with unawareness of HIV-positive status in bivariate log binomial regression, for example certain regions and primary education. It’s possible that these were not included due to a correction for multiple hypothesis testing but it could be helpful to clarify why those positive results in the table are not included in the text.

Line 225: It may help the reader to clarify that the Crude and Adjusted Prevalence Ratios are estimated using a log binomial model. For less statistically-inclined readers (including myself), there are a lot of terms in the methods that could be confusing.

Line 230: Why were these three variables controlled for? They were not the “priority confounders” as stated in line 121, nor are they significant variables in Table 2.

Line 232: the word “statistically” is unnecessary here as it already says “significantly”

Line 233: Previously in the Results section, 95% CI were followed by a colon, but in these next two paragraphs, they are followed by a comma in some cases, colon in others, and no punctuation as well.

Discussion:

Line 246: one fifth should be “one-fifth” – applies throughout the paper.

Line 263: “Kuwazulu Natal” should be “KwaZulu-Natal province”

Line 287: There is a typo in “2030These and measures”

Line 292: “door to door” should be hyphenated

Line 296: “contribute for the” should be “contribute to the”

Line 317: “Voluntary Medical ale Circumcision” should be “Voluntary Medical Male Circumcision”

Line 318: There is an extra period

Line 327: “HIV positives participants” should be “HIV positive participants”

Line 350: “A previous study report indicated” should read “A previous study reported”

Line 353: “have a steady partner, and are not part” should read “have a steady partner, are not part”

Limitations:

Line 371: “Furthermore;” should read “Furthermore,”

Conclusions:

Line 379: There is an extra period at the end

I would add as one of the major conclusions that the number of unaware HIV+ individuals has a different distribution than the HIV prevalence or % unaware, which is a critical distinction to control the epidemic.

Discretionary Revisions

Abstract:

Lines 26-33: Could probably be shortened to something along the lines of: “In the bivariate analysis…, three regions (Oromia, Addis Ababa, and Harari), male gender, and young age (15-24 years) were significantly associated with HIV awareness of HIV status. In multivariate analysis, the same variables were associated with awareness of HIV status.”

Line 31: The Adjusted Prevalence Ratios are actually risks and not odds. (Unless a prevalence odds ratio was used, which doesn’t appear to be the case)

Introduction:

Lines 42-49: This paragraph jumps from discussing mortality (“mortality declined by 44%”) to infections (“infections decreased by 28%”), and then back to mortality (“The global decline in deaths from AIDS-related illness”). I would consider centering the introduction on the UNAIDS 2020 targets regarding awareness of HIV status.

Methods:

Line 116: May want to describe Jackknife replication. I’m unsure if it’s commonly used but readers may be more familiar with bootstrapping.

Results:

Line 144: A table of characteristics of the general sample, including both HIV-positive and HIV-negative could be useful for comparison’s sake.

Lines 181-184: This sentence is a little unclear. May want to consider changing the wording to something along the lines of “In Afar, Addis Ababa, and Dire Dawa, more people never tested than self-reported negative, whereas in the rest of the other regions, more people self-reported negative.”

Line 206: Figure 3 seems to show in graphical representation the data in the first portion of Table 2. Is there a particular reason for the geographic representation of the unawareness? If not, can consider removing this figure.

Discussion:

Line 259: Please explain the overall capacity score.

Line 268: Just a note, this is exactly why the reader would be very interested in the ARV data!

Line 298: Although this is true regarding Gambella, as you stated in line 283, this would be a low-yield strategy.

Line 308: The authors discuss the difference in transmission between PLHIV who are aware and unaware of their HIV status. They should consider a modeling analysis in the future!

6. PLOS authors have the option to publish the peer review history of their article (what does this mean?). If published, this will include your full peer review and any attached files.

Reviewer #1: **Yes: **Roger Ying

---

## [Author Response · Author response to Decision Letter 0]

17 Jun 2021

Point-by-point response to the editor and reviewer is uploaded.

---

## [Decision Letter · Decision Letter 1]

12 Jul 2021

Factors associated with Unawareness of HIV Positive Status in Urban Ethiopia: Evidence from the Ethiopia Population-based HIV Impact Assessment 2017-2018

PONE-D-21-05302R1

Dear Dr. Lulseged,

We’re pleased to inform you that your manuscript has been judged scientifically suitable for publication and will be formally accepted for publication once it meets all outstanding technical requirements.

Kind regards,

José Antonio Ortega, Ph.D.

Academic Editor

PLOS ONE

Additional Editor Comments (optional):

Both the editor and the reviewer comments have been satisfactorily addressed.

Further attention should be paid to whether the data sharing statement is agreeable to the journal policy. 

Reviewers' comments:

Reviewer's Responses to Questions

**Comments to the Author**

1. If the authors have adequately addressed your comments raised in a previous round of review and you feel that this manuscript is now acceptable for publication, you may indicate that here to bypass the “Comments to the Author” section, enter your conflict of interest statement in the “Confidential to Editor” section, and submit your "Accept" recommendation.

Reviewer #1: All comments have been addressed

2. Is the manuscript technically sound, and do the data support the conclusions?

Reviewer #1: Yes

3. Has the statistical analysis been performed appropriately and rigorously? 

Reviewer #1: Yes

4. Have the authors made all data underlying the findings in their manuscript fully available?

Reviewer #1: Yes

5. Is the manuscript presented in an intelligible fashion and written in standard English?

Reviewer #1: Yes

6. Review Comments to the Author

Reviewer #1: Thank you for addressing my prior concerns. My remaining comment is that the Discussion is quite long and unfocused. I have two suggestions for shortening. One is that comparisons to other studies can be shortened. For example, lines 255-256 are well summarized whereas lines 330-336 could be shortened without sacrificing information. The second suggestion is that the discussion can be more targeted. The discussion begins with a comparison of the unawareness rate in Ethiopia compared to other countries, and then hypothesizes several reasons why this is so. It then suggests why unawareness may be more important than prevalence. It then switches to geographic variations in testing strategies, then consequences of unawareness, geographic variation again, and then risky sexual activity. Although all sections are relevant and important, the reader would benefit from guidance regarding the most salient results rather than all.

7. PLOS authors have the option to publish the peer review history of their article (what does this mean?). If published, this will include your full peer review and any attached files.

Reviewer #1: **Yes: **Roger Ying

---

## [Editor Report · Acceptance letter]

30 Jul 2021

PONE-D-21-05302R1 

Factors associated with unawareness of HIV-positive status in urban Ethiopia: Evidence from the Ethiopia Population-based HIV Impact Assessment 2017-2018 

Dear Dr. Lulseged:

I'm pleased to inform you that your manuscript has been deemed suitable for publication in PLOS ONE. Congratulations! Your manuscript is now with our production department. 

Kind regards, 

on behalf of

Dr. José Antonio Ortega 

Academic Editor

PLOS ONE